# This Is Not What We Ordered: Exploring Why Biased Search Result Rankings Affect User Attitudes on Debated Topics

Tim Draws
Delft University of Technology
t.a.draws@tudelft.nl

Nava Tintarev
Maastricht University
n.tintarev@maastrichtuniversity.nl

Ujwal Gadiraju
Delft University of Technology
u.k.gadiraju@tudelft.nl

Alessandro Bozzon
Delft University of Technology
a.bozzon@tudelft.nl

Benjamin Timmermans
IBM
b.timmermans@nl.ibm.com

## ABSTRACT

In web search on debated topics, algorithmic and cognitive biases strongly influence how users consume and process information. Recent research has shown that this can lead to a *search engine manipulation effect* (SEME): when search result rankings are biased towards a particular viewpoint, users tend to adopt this favored viewpoint. To better understand the mechanisms underlying SEME, we present a pre-registered, $5 \times 3$ factorial user study investigating whether *order effects* (i.e., users adopting the viewpoint pertaining to higher-ranked documents) can cause SEME. For five different debated topics, we evaluated attitude change after exposing participants with mild pre-existing attitudes to search results that were overall viewpoint-balanced but reflected one of three levels of algorithmic ranking bias. We found that attitude change did not differ across levels of ranking bias and did not vary based on individual user differences. Our results thus suggest that order effects may not be an underlying mechanism of SEME. Exploratory analyses lend support to the presence of *exposure effects* (i.e., users adopting the majority viewpoint among the results they examine) as a contributing factor to users' attitude change. We discuss how our findings can inform the design of user bias mitigation strategies.

## CCS CONCEPTS

• **Information systems → Web searching and information discovery**; • **Human-centered computing → User studies**.

## KEYWORDS

Web Search; Ranking Bias; User Attitudes; User-centered Evaluation

**ACM Reference Format:**
Tim Draws, Nava Tintarev, Ujwal Gadiraju, Alessandro Bozzon, and Benjamin Timmermans. 2021. This Is Not What We Ordered: Exploring Why Biased Search Result Rankings Affect User Attitudes on Debated Topics. In *Proceedings of the 44th International ACM SIGIR Conference on Research and Development in Information Retrieval (SIGIR '21), July 11–15, 2021, Virtual Event, Canada.* ACM, New York, NY, USA, 11 pages. https://doi.org/10.1145/3404835.3462851

## 1 INTRODUCTION

Imagine that, in an effort to decide on future policies, the government is seeking informed opinions from the population concerning whether zoos should (continue to) exist. You happen to be one of the randomly selected individuals asked for such an informed opinion but you currently have a mild (i.e., uncertain) attitude towards zoos. Intending to build a strong, informed opinion by exposing yourself to different existing viewpoints on the topic, you enter the query "should zoos exist?" into a web search engine.

Web search on such debated topics may be biased in several different ways [7]. Next to algorithmic ranking biases that cause documents pertaining to certain beliefs to be ranked higher than others [18, 51, 58–60], cognitive user biases such as the *confirmation bias* strongly affect how users consume and process information from search results [6]. A well-established finding is that users typically pay more attention to higher-ranked items when consuming ranked lists of search results [27]. This phenomenon – known as *position bias* – leads users to click on results at higher ranks with greater probability [27, 29, 44] and primarily engage with the first *search engine results page* (SERP) [20]. Consequently, the ranking of search results greatly influences users' post-search attitudes when exploring debated topics (e.g., whether zoos should exist): recent work has demonstrated that when a search result ranking is biased towards any particular viewpoint (i.e., assigning higher ranks to documents that express it), users tend to change their attitude accordingly [3, 15, 47]. This type of attitude change due to viewing a biased ranked list of search results has been called the *search engine manipulation effect* (SEME) [15]. It can occur even after single search sessions, for a variety of topics (e.g., political elections and medical treatment) [3, 15, 47], and without users' awareness [19].

Why do users fall prey to SEME? Although position bias can explain how users *select* search results to engage with, it does not explain how users *process* the search results they have picked for consumption. Two different cognitive biases have been suspected to drive the information processing that leads to SEME: *exposure effects* and *order effects* [6]. Exposure effects imply that being exposed to messages pertaining to a particular viewpoint increases an individuals' favorability towards that viewpoint [6, 38, 66]. In the context of web search, this would mean that users' tendency to adopt a particular viewpoint increases with the proportion of consumed documents that express this viewpoint. Order effects occur when users assign *more weight* to information drawn from higher-ranked results [6]. This would mean that the influence of a document's expressed viewpoint is weighted by its rank.

Despite these considerations, there is a lack of empirical evidence as to whether exposure effects, order effects, or both are responsible for SEME. Previous studies have demonstrated SEME using search result lists that had a majority viewpoint among documents on the first SERP [3, 15, 47, 60], which makes both exposure effects and order effects plausible (but not necessary) explanations. For instance, suppose a user queries "should zoos exist?" and sees eight documents supporting zoos and two documents opposing zoos on the first SERP. Assuming that the user engages with only these first ten search results, they may change their attitude towards a favorability for zoos because, among the results they consumed, zoo-supporting documents were in the majority (i.e., exposure effects) and ranked higher (i.e., order effects). More recent research indicates that users may be looking for majority viewpoints but are unaware of any order effects when they search the web [19]. However, humans are often unaware of their biases [50], so it is currently unclear which cognitive processes truly contribute to SEME.

Mitigating SEME requires a thorough understanding of its underlying mechanisms. This research investigates whether cognitive order effects contribute to SEME by studying the influence of algorithmic ranking bias for *overall viewpoint-balanced* top 10 search results (i.e., the first SERP) on user attitudes towards debated topics. Unlike previous research, this method exposes users to SERPs that contain equal proportions of opposing and supporting documents, thereby mitigating potential exposure effects and isolating potential order effects. To explore which users might be particularly vulnerable to SEME, we additionally study whether factors such as *actively open-minded thinking*, *user engagement*, and *perceived diversity* play a role here. We have four research questions:

- **RQ1.** Does the ranking of overall viewpoint-balanced top 10 search results affect attitude change concerning debated topics in users with mild pre-existing attitudes?
- **RQ2.** Are individual user characteristics such as *actively open-minded thinking* and *user engagement* related to attitude change?
- **RQ3.** Do factors such as *actively open-minded thinking*, *user engagement*, or *perceived diversity* interact with search result rankings to cause attitude change?
- **RQ4.** Are users aware of varying degrees of viewpoint diversity in the search results they consume?

In sum, through this work we make the following contributions:

- We release a viewpoint-annotated data set containing real search results on five different debated topics (see Section 4).
- We present a pre-registered, $5 \times 3$ factorial user study exploring how biased search result rankings may cause attitude change.[1]

We find that most users changed their attitude as a result of viewing the search results. However, attitude change did not differ across different result rankings, a finding that contravenes the predictions of order effects. We similarly find no evidence that individual differences (i.e., actively open-minded thinking, user engagement) or perceived diversity affect attitude change directly or in interaction with search result rankings. Exploratory analyses suggest that *exposure effects* may explain attitude change.

The preregistration, data sets, user study materials, results, and data analysis code related to this research are openly available.[2]

## 2 RELATED WORK AND HYPOTHESES

In this section, we discuss related work and present our hypotheses, which had all been preregistered before any data collection.

### 2.1 Cognitive Biases in Web Search

The order in which search results are ranked strongly affects user preferences and behavior [22, 27–29, 41, 43, 44]. Users exhibit a *position bias* when exploring search results: they tend to pay more attention to results at higher ranks [44] and are more likely to click and examine them [27, 29, 43]. As a result, users usually do not even examine search results beyond the first result page [20].

More recent research has shown that this preference for consuming higher-ranked search results can affect attitudes of users with mild pre-existing opinions [3, 15, 47, 60]. Specifically, the findings of these studies suggest that when documents that express a particular viewpoint are systematically ranked higher than other documents (i.e., when there is an algorithmic, viewpoint-related *ranking bias*), users tend to adopt this more prominent viewpoint. This effect (i.e., SEME) is particularly prominent when search results are biased towards a *supporting viewpoint* (e.g., supporting the idea that zoos should exist) [60]. It can occur even after single search sessions, for a variety of topics (e.g., political elections, medical treatment, and vaccinations) [3, 15, 47, 60], and without users' awareness [19]. Given that real search result rankings are often biased concerning viewpoints [18, 51, 58–60], SEME is a pressing concern [7, 9].

Azzopardi [6] argues that two well-known cognitive biases may contribute to SEME: *exposure effects* and *order effects*. Exposure effects occur when exposure to messages pertaining to a particular viewpoint increases an individuals' favorability towards that viewpoint [38, 66]. In the context of web search, this would mean that the higher the proportion of consumed documents expressing a particular viewpoint, the greater user's tendency of adopting that viewpoint. Order effects imply a *weighting* of consumed information according to its position in a given order (e.g., assigning more weight to information encountered first) [24]. Order effects may nudge users to adopt the viewpoint that is expressed by highly-ranked documents – even when they consume an overall viewpoint-balanced set of search results that has no majority viewpoint.[3]

Both exposure effects and order effects are plausible explanations for SEME but there is currently a lack of empirical understanding as to whether either of them (or both) are responsible for SEME. In previous studies that explored SEME, the first SERP that participants saw reflected a viewpoint imbalance so that one viewpoint was in the majority among the results on the first page [3, 15, 47, 60]. This means that either exposure effects (i.e., users adopting the majority viewpoint among the results they consume), order effects (i.e, users adopting the viewpoint of higher-ranked results), or a combination of both could have been responsible for SEME. For example, White and Horvitz [60] demonstrated SEME by comparing imbalanced, ranking-biased SERPs (i.e., allowing for both exposure and order

---

[1]Pre-registering our study meant publicly announcing the hypotheses, experimental setup, and analysis plan we describe in this paper before data collection. We have adjusted some hypothesis descriptions for clarity (e.g., H1a). Our (time-stamped) preregistration is available at https://osf.io/qd27x (anonymized for blind peer-review).

[2]A repository with supplementary material can be found at https://osf.io/6tbvw/.
[3]Order effects could also mean weighting in favor of information encountered *last*. However, in situations such as web search, where the number of consumed items is typically low [44], assigning more weight to information seen *first* is more likely [6, 24].

effects to take place) to a controlled setting in which the shown SERPs were viewpoint-balanced *and* ranked in random order (i.e., ruling out both exposure and order effects). Recent research indicates that exposure effects may indeed underlie SEME [19] but the evidence surrounding order effects in this context is inconclusive. Users put more *trust* in higher-ranked results [27] and it has been argued that the viewpoint expressed by the first search result acts as an *anchor* in users' exploration of search results [41]. However, users do not consciously experience order effects [19], which have – to the best of our knowledge – not been demonstrated in situations where potential exposure effects are mitigated or ruled out.

If order effects underlie SEME, users will tend to adopt the viewpoint expressed by higher-ranked results, at least when they have mild pre-existing attitudes [15, 60]. SEME should then occur even when users consume a ranking-biased but overall viewpoint-balanced set of documents. Our first hypothesis follows earlier work arguing that order effects are (partly) responsible for SEME [6, 41].

**Hypothesis 1a (H1a).** *The ranking of an overall viewpoint-balanced list of 10 search results affects attitude change towards debated topics in users with mild pre-existing attitudes.*

Although SEME has been demonstrated for a variety of topics, to the best of our knowledge, no previous research has assessed topical differences concerning SEME. Given that some topics are more polarizing than others, we expect that the effect of search result rankings on attitude change is moderated by topic.

**Hypothesis 1b (H1b).** *Topic moderates the effect of search result rankings on attitude change.*

## 2.2 Vulnerability of Users to Attitude Change

Users with mild pre-existing attitudes are more susceptible to attitude change when searching the web compared to users with strong opinions [60]. If biased search result rankings can cause attitude change in such users (i.e., elicit SEME), an important step towards developing mitigation strategies is to understand which other factors (aside from having a mild pre-existing attitude) characterize users who are particularly affected. Psychological research has identified that willingness to process (counter-attitudinal) information [55] and engagement with the topic at hand [46] may increase an individual's vulnerability to attitude change.

We thereby defined two distinct user-specific factors that we expected to (1) predict attitude change directly and (2) affect attitude change in interaction with search result rankings. First, we predicted such a role for *actively open-minded thinking* (AOT). AOT is a style of thinking that involves considering counter-attitudinal information and opinions of others when forming one's own opinion [23]. Consequently, AOT predicts information acquisition [23] and reasoning independently from pre-existing attitude [54].

**Hypothesis 2a (H2a).** *Actively open-minded thinking (AOT) predicts attitude change in web search.*

**Hypothesis 3a (H3a).** *Actively open-minded thinking (AOT) moderates the effect of search result rankings on attitude change.*

Second, we hypothesized that *user engagement* will act as a direct and moderating factor in this context. High interaction with a search result list may be analogous to strong engagement with a topic. Moreover, depending on the degree of ranking bias present

in a search result list, engaged users may be exposed to a growing diversity of viewpoints as they move down the search results list.

**Hypothesis 2b (H2b).** *User engagement predicts attitude change in web search.*

**Hypothesis 3b (H3b).** *User engagement moderates the effect of search result rankings on attitude change.*

It has been shown that higher engagement with presented information mediates the relationship between AOT and task performance [23], which is why we expected the same in our study.

**Hypothesis 2c (H2c).** *User engagement mediates the relationship between AOT and attitude change.*

Additionally, we expected perceived diversity to moderate the effect of search result rankings on attitude change. Perceiving search result lists as more or less diverse could reflect the degree to which users have considered the different viewpoints present in the topic. Merely perceiving that a search result list has a high diversity could therefore change how a search result ranking affect attitude change.

**Hypothesis 3c (H3c).** *Perceived diversity in search result lists moderates the effect of search result rankings on attitude change.*

## 2.3 User Perception of Viewpoint Diversity

To the best of our knowledge, no previous work has explored whether users perceive an existing (lack of) viewpoint diversity in sets of search results. We investigate the effect of search result rankings on perceived diversity to better understand how and why user attitudes might be affected. Because previous research has shown that users truly engage with only the top few results on a SERP [27, 43, 44], we expected that the ranking of search results would skew their perception of diversity in the search results.

**Hypothesis 4a (H4a).** *Search result rankings affect perceived diversity in search result lists.*

## 2.4 Measuring Ranking Bias in Search Results

Much research has been devoted to measuring *diversity* in search result rankings [1, 2, 13]. These methods usually reward both *diversification* (i.e., absence of bias) and *relevance* of search results across a ranking. Doing so, they aim to maximize the *utility* of search results. However, a trade-off with document relevance is not desired when measuring *viewpoint-related ranking bias*, as the ultimate aim here is not to maximize utility for the user but to mitigate biases.

Draws et al. [14] suggest using *ranking fairness metrics* to assess viewpoint-related ranking bias in search results. They classify documents into viewpoint categories; i.e., into ordinal categories ranging from *strongly opposing* to *strongly supporting* a claim. Fair ranking typically aims for *statistical parity*: a setting where a pre-defined *protected attribute* (e.g., expressing a particular viewpoint) does not influence a document's position in the ranking. By evaluating statistical parity for different top portions of a ranking and applying a discount function to account for the decrease in user attention as ranks go up, ranking fairness metrics can assess how fairly represented different viewpoints are across a ranking. Several methods have been proposed for measuring ranking fairness [8, 53, 61, 63], including *normalized discounted difference* (nDD) and *normalized discounted Kullback-Leibler divergence* (nDKL) [14, 63].

Kulshrestha et al. [34] propose *ranking bias* (RB), a metric that assesses the degree to which one side of an issue is overly represented in a ranking. RB considers *continuous* viewpoint labels; e.g., documents could be labeled on a continuous scale ranging from −3 (strongly opposing) to 3 (strongly supporting). RB computes the mean viewpoint label for different top portions of the ranking and compares them with the overall mean. These comparisons are then aggregated in a discounted fashion.

## 3 PRELIMINARIES

Table 1 shows the taxonomy of viewpoint labels used in this paper. Given a claim (e.g., *zoos should exist*), a document can be classified into one of seven different categories based on its position with respect to the claim. This is analogous to rating a document on a seven-point Likert scale ranging from *extremely opposing* to *extremely supporting*. The viewpoint labels can be represented as integers ranging from −3 to 3, where negative values indicate an *opposing viewpoint*, 0 indicates a *neutral viewpoint*, and positive values indicate a *supporting viewpoint* towards the claim.

**Table 1: The viewpoint label taxonomy we considered.**

| Viewpoint Label | | Example (Topic: *Should Zoos Exist?*) |
|---|---|---|
| -3 | strongly opposing | "Horrible places! All zoos should be closed." |
| -2 | opposing | "We should strive towards closing all zoos." |
| -1 | somewhat opposing | "Despite some benefits, I'm against zoos." |
| 0 | neutral | "We present arguments for and against zoos." |
| +1 | somewhat supporting | "Zoos are not great, but they benefit society." |
| +2 | supporting | "I'm in favor of zoos, let's keep them." |
| +3 | strongly supporting | "There is nothing wrong with zoos!" |

We align the notion of viewpoint diversity in search results with three different offline metrics. First, we consider two different fairness metrics that can evaluate the degree to which a predefined *protected viewpoint* (e.g., the opposing or supporting viewpoint) is advantaged or disadvantaged across a ranked list: nDD and nDKL [14, 63]. These metrics operate by iteratively either comparing the proportion of protected viewpoints across the ranking with the overall proportion (nDD) or computing the *Kullback-Leibler divergence* [33] between the binomial distributions of protected and non-protected viewpoints across the ranking and overall (nDKL). Second, we consider RB, which assesses the degree to which a search result ranking favors a particular side of a debated topic by taking into account document viewpoint labels from a scale (e.g., −3 to 3) [34]. In line with earlier work, we discount the metric computation logarithmically in steps of one to incorporate the notion that higher-ranked results receive more attention [14]. The code that implements them can be found on our repository.

## 4 DATA SET

### 4.1 Debated Topics

We first conducted a preliminary study to identify a set of disputed topics that most people hold undecided or mild viewpoints on (i.e., because we aimed to test users with mild viewpoints). We picked 18 different topics from *ProCon*, a website that lists controversial issues [48].[4] We then created a survey in which participants could

state their opinion on each of the 18 topics using a seven-point Likert scale ranging from "strongly agree" to "strongly disagree". Each topic was phrased as a question (e.g., "Should zoos exist?"). A total of 100 participants completed the survey for a $0.60 reward after being recruited from the online participant pool *Prolific* [49]. We excluded seven responses from data analysis due to failing at least one of two attention checks that we had included.

We defined two inclusion criteria for topics. First, we aimed to include topics for which attitudes were generally not skewed towards a particular side. We evaluated this by transforming all survey responses to integers ranging from −3 (i.e., "strongly disagree") to 3 (i.e., "strongly agree") and subsequently conducting one-sample Wilcoxon tests against a test value of 0 for each topic. A significant result in this test suggested that the mean attitude on the topic at hand is not undecided (i.e., not equal to 0). We thus included only topics for which the Wilcoxon test had a *non-significant* result.[5] Second, we desired topics that a majority of people held a mild (i.e., uncertain) attitude towards. We implemented this criterion by classifying all survey responses into *mild* and *strong* viewpoints: responses among the three central options from the Likert scale (i.e., ranging from "somewhat disagree" to "somewhat agree") were mapped onto the *mild* class, all other responses were mapped to the *strong* class. We included a topic in our study only if the proportion of mild attitudes was above 0.5. Five topics met both criteria and were therefore included in our study:

(1) *Are social networking sites good for our society?*
(2) *Should zoos exist?*
(3) *Is cell phone radiation safe?*
(4) *Should bottled water be banned?*
(5) *Is obesity a disease?*

### 4.2 Search Results

Per topic, we created a set of 14 queries according to a pre-defined template. This template included neutrally-formulated queries (e.g., "zoos opinions", "zoos arguments") as well as viewpoint-biased queries (e.g., "opinions supporting zoos", "arguments opposing zoos").[6] We then retrieved the top 50 search results for each of these queries using the API of the search engine *Bing* [37].

From the search results we retrieved on each topic, we handpicked 56 opinionated search result items and had them annotated by crowd workers on *Amazon Mechanical Turk* [4]. We collected at least three annotations per item; both for relevance (binary) and viewpoint concerning the topic at hand (seven-point Likert scale from "strongly opposing" to "strongly supporting"). We paid crowd workers $2 per task in which they annotated 14 different search results. Additionally, workers could earn a $0.50 bonus if they passed two attention checks. Data from participants who did not pass at least one attention check were excluded from analysis. According to Krippendorff's $\alpha$, inter-rater reliability for the viewpoint judgments was satisfactory ($\alpha$ = 0.79) [32]. Qualitative feedback from crowd workers revealed that the task was understandable and could be performed without issues. We assigned each search result item the median annotation for both of these measurements.

---

[4]We excluded topics that were highly politicized (e.g., *gun control* or *abortion*).

[5]We corrected for multiple testing by applying a Bonferroni correction; i.e., only $p$-values below $\frac{0.05}{18}$ = 0.003 were considered significant.

[6]The full list of queries we used is available on our repository; see Footnote 2.

**Table 2: Three conditions representing the three levels of ranking bias. Here, all rankings are biased towards the opposing viewpoint, but our study also included their symmetrical opposites (favoring the supporting viewpoints).**

| | Viewpoint label | | |
|---|---|---|---|
| **Rank** | *Little bias* | *Moderate bias* | *Extreme bias* |
| 1 | -1 | -2 | -2 |
| 2 | 2 | -2 | -2 |
| 3 | 1 | 1 | -2 |
| 4 | -2 | 2 | -1 |
| 5 | -1 | -1 | -1 |
| 6 | 2 | 2 | 1 |
| 7 | -2 | -2 | 1 |
| 8 | 2 | 2 | 2 |
| 9 | -2 | -1 | 2 |
| 10 | 1 | 1 | 2 |

Our final data set thus consisted of 280 search result items (including name, snippet, and URL) that were annotated concerning their relevance and viewpoint with respect to the five debated topics.

## 5 METHOD AND EXPERIMENTAL SETUP

This section describes the materials, procedure, participants, and statistical analysis related to our user study. Next to constructs introduced in Section 2, we here describe several additional measurements that we included in our study (i.e., topical interest, gender, and age). We used these measurements for descriptive and exploratory analyses; more specifically, to obtain a clearer image of our sample (e.g., whether participants had a realistic level of topical interest) and to explore directions for future research.

### 5.1 Materials

*5.1.1 Search Result Rankings.* Using the data set described in Section 4, we assembled one set of ten search results for each of the five topics. We did that by randomly sampling three "opposing", two "somewhat opposing", two "somewhat supporting", and three "supporting" items from the search result items that were deemed relevant to a given topic by crowd workers.[7] Thus, although the search result lists were different in terms of topic and content, they were consistent with respect to the representation of viewpoints.

We ranked the search result sets to reflect three levels of bias (*little*, *moderate*, and *extreme*) by computing viewpoint diversity for all possible ranking orders using the three metrics introduced in Section 3, and summing up the metric outcomes for each ranking. We selected ranking permutations for each level based on their score; *little bias*– lowest combined score, *moderate bias*– closest to the mean, and *extreme bias*– highest combined score.

Table 2 illustrates the three conditions with a bias toward opposing viewpoints. In practice, we counter-balanced the search result rankings so that half contained bias for the opposing viewpoint and half the supporting viewpoint.[8]

*5.1.2 Actively Open-Minded Thinking (AOT) scale.* A 7-item scale that measures the degree to which a person is willing to consider

---

[7]We did not include "strongly opposing" and "strongly supporting" items in the search result sets because they were non-existent for several topics.

[8]Note that the metrics have the same output for symmetrical search result rankings (e.g., ranking all opposing documents before any supporting documents and vice versa).

opposing viewpoints and change their mind about topics [23]. Responses were recorded on a seven-point Likert scale ranging from "strongly agree" to "strongly disagree" and later aggregated by taking their mean. To ensure reliability of responses, an attention check item was added to the AOT scale.

*5.1.3 User Engagement Scale - Short Form (UES-SF).* A 12-item scale that measures the degree to which a person was involved and satisfied with a given experience [42]. Responses were recorded on a seven-point Likert scale and averaged.

*5.1.4 Perceived diversity scale.* We measured *perceived diversity* using adapted versions of items from a scale for measuring recommendation variety in a list of recommended items [30]. Responses were recorded on a seven-point Likert scale and averaged.

### 5.2 Variables

**Independent variables**

- *Topic* (categorical; between-subjects). Each participant saw search results that relate to one of five different debated topics that we included in this study (see Section 4.1).
- *Condition* (categorical; between-subjects). Each participant was randomly assigned to one of three conditions that each involve a different ranking of search results. These search result rankings reflected (1) little, (2) moderate, and (3) extreme ranking bias (see Section 5.1.1). This variable was nested within *topic*.

**Dependent variable**

- *Attitude change* (continuous). We measured each participant's attitude towards their assigned topic twice on a seven-point Likert scale (i.e., once before and once after exposing them to a ranked list of search results related to a debated topic). Similar to previous research [15], we computed *attitude change* by subtracting the first measurement from the second: [-6,6]. Additionally, we computed the *absolute attitude change*: [0,6].

**Covariates**

- *Actively open-minded thinking* (continuous). Measured using the AOT scale: [0;6].
- *User engagement* (continuous). We quantified user engagement by aggregating three different metrics: UES-SF, total time spent examining the search results, and the number of links that a user clicked. We normalized each of these metrics and then aggregated them by taking their mean: [0;1].
- *Perceived diversity* (continuous). We measured the degree of viewpoint diversity that participants perceived in the search results using the perceived diversity scale: [0;6].

**Descriptive and exploratory measurements**

- *Gender.* Participants could select their self-identified gender.
- *Age.* Participants could type their age in an open text field.
- *Topical interest* (continuous). We measured interest in the topic at hand using the item "I was interested in learning more about this topic", which could be answered by selecting the appropriate option from a seven-point Likert scale: [0,6].

### 5.3 Procedure

We conducted our study on the online task platform *Qualtrics* [52]. The procedure had been approved by the research ethics committee

at our institution. After agreeing to an informed consent, participants went through three subsequent steps:

*Step 1.* Participants received a short introduction to the task and subsequently stated their gender, age, and attitude towards each of the five debated topics. The introduction read:

"*Imagine the government is seeking informed opinions from the population related to a number of debated topics. In order to decide on future policies, they would like to know what the public thinks. You happen to be one of the randomly selected individuals the government is asking for such an informed opinion.*"

*Step 2.* Participants were assigned to one of the topics they held a *mild* viewpoint on (i.e., responding with "somewhat agree", "neither agree nor disagree", or "somewhat disagree").[9] They learned which topic they had been assigned to and were instructed to pick one of 14 different queries for their web search.[10] Here, the sole purpose of selecting a query was to make the task more realistic. Participants did not get different treatment based on the query they picked.

*Step 3.* Participants were randomly assigned to one of the three conditions and were presented with a list of search results. This list contained search result items that were relevant to the assigned topic and ranked according to the assigned condition. For example, if a participant was assigned the topic "Should zoos exist?" and the condition *extreme bias*, that participant saw a search result list in which all documents supporting zoos were ranked above all documents opposing zoos (or vice versa). At the bottom of the page was a "more" button that participants could click but that would not yield them more search results. This allowed us to study the number of participants who might have explored further results if they were available. Participants could explore the search results by reading the names and snippets, or by directly examining the web pages they found most interesting. They had to spend at least two minutes exploring the search results but otherwise could take as much time as they need for this part of the study.

*Step 4.* Participants stated their (updated) attitude and interest concerning their assigned topic.

*Step 5.* Participants filled in a post-questionnaire that consisted of the AOT scale, the UES-SF, and the perceived diversity scale.

### 5.4 Statistical Analyses

We performed an *Analysis of Covariance* (ANCOVA) using *absolute attitude change* as dependent variable, *condition* and *topic* as between-subjects factors and *AOT*, *user engagement*, and *perceived diversity* as covariates.[11] We looked at the main effects of *condition* (H1a), *AOT* (H2a), and *user engagement* (H2b) on *attitude change* as well as the interaction effects of *condition* and *topic* (H1b), *condition* and *AOT* (H3a), *condition* and *user engagement* (H3b), and *condition* and *perceived diversity* (H3c). Additionally, we conducted

a One-Way *Analysis of Variance* ANOVA to analyze the main effect of *condition* on *perceived diversity* (H4a). We decided to conduct AN(C)OVAs despite the anticipation that our data may not be normally distributed because these analyses have been shown to be robust to Likert-type ordinal data [40]. To correct for testing nine hypotheses, we applied a Bonferroni correction so that the significance threshold decreased to $\frac{0.05}{9} = 0.006$.

We also conducted two Bayesian ANOVAs according to the protocol proposed by van den Bergh et al. [56]. In contrast to classical (frequentist) analyses, Bayesian hypothesis tests quantify evidence that the data provide *in favor of the null hypothesis* as opposed to the alternative hypothesis [57]. This is especially useful when trying to interpret non-significant results from classical hypothesis tests because such results do not mean that the null hypothesis is true [21]. Practically, performing Bayesian hypothesis tests allowed us to weigh the evidence in favor of some of the null hypotheses opposing the hypotheses laid out in Section 2. We performed these analyses using the software JASP [25] with default settings. We computed *Bayes Factors* (BFs) by comparing the model of interest to a *null model*[12] and interpret them in adherence to the guide proposed by Lee and Wagenmakers [35], who adopted it from Jeffreys [26].

### 5.5 Participants

Before recruiting participants, we computed the required sample size in a power analysis for a *Between-Subjects ANOVA* using the software *G\*Power* [17]. We specified the default effect size $f = 0.25$ (i.e., indicating a moderate effect), a significance threshold $\alpha = \frac{0.05}{9} = 0.006$ (i.e., due to testing multiple hypotheses; see Section 5.4), a statistical power of $(1-\beta) = 0.8$, and that we will test $5 \times 3 = 15$ groups (i.e., three conditions, five topics). We computed the required sample size for each of our hypotheses using their respective degrees of freedom. This resulted in a required sample size of 368 participants.

We thus recruited 391 participants from *Prolific* (reward: \$2). All participants were proficient English speakers above the age of 18. We excluded participants from data analysis if they did not hold a mild viewpoint towards any of the five topics, failed at least one attention check, or represented an outlier in terms of the amount of time they spent exploring the SERP. Outliers were participants (seven in total) who spent more or less time on the SERP than two standard deviations from the mean time spent.[13] The resulting sample of 364 participants had an average age of 37 (sd = 13) and a balanced gender distribution (59% female, 41% male, < 1% other).

## 6 RESULTS

In this section, we present the results of our study. We discuss descriptive statistics, the outcomes of the hypothesis tests we conducted, and exploratory findings.

### 6.1 Descriptive Statistics

Participants were distributed over the five topics as follows: 25 (*social networking sites*), 9 (*zoos*), 48 (*cell phone radiation*), 73 (*bottled water*), and 209 (*obesity*).[14] Whereas most participants (81.6%)

---

[9]Participants without a mild viewpoint on any topic were ejected from the study.
[10]The queries that participants could choose from were the same 14 queries that we used for retrieving the search results per topic (see Section 4).
[11]We made two adjustments here compared to our preregistration. First, our preregistration stated that we would perform an ANOVA using these variables; however, ANCOVA is the suitable analysis. Second, we used *absolute* (instead of raw) *attitude change* as dependent variable. SERPs favored either the supporting or opposing viewpoint, which means that raw scores could have balanced each other out. We were chiefly interested in the *magnitude* of attitude change here.

[12]Here, the *null model* contained nothing but an intercept.
[13]This exclusion criterion was not mentioned in our preregistration, but we felt it was necessary as some participants spent excessive amounts of time on the SERP.
[14]Note that random (balanced) allocation of participants over topics was not possible because we specifically targeted users with mild pre-existing attitudes.

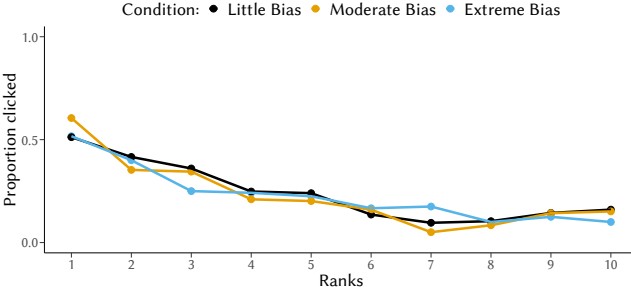

**Figure 1: Click proportions over the ranks. Users behaved similarly across conditions, showing a weak position bias.**

chose neutral queries (e.g., "is obesity a disease?") for their task, some picked either opposing queries (6.6%; e.g., "why cell phone radiation is unsafe") or supporting queries (11.8%; e.g., "arguments supporting zoos"). The number of participants was balanced between conditions: there were 125, 119, and 120 participants in the *little bias*, *moderate bias*, and *extreme bias* conditions, respectively. Most participants (i.e., 83%) were at least somewhat interested in their assigned topic. Overall AOT (mean = 3.89, sd = 0.41), user engagement (mean = 0.33, sd = 0.10), and perceived diversity (mean = 3.40, sd = 0.94) were moderate.

Figure 1 shows the proportion of clicks over the ranks for each of the three conditions. In all conditions, we observe that click proportions decrease from roughly 0.55 at the first rank to roughly 0.15 at the sixth rank and below. This reflects a weaker position bias compared to what previous research has found, where click proportions decrease much more severely with the ranks (i.e., from similar proportions at the first rank down to approximately 0.03 at the tenth rank) [27, 43, 44]. As expected, it thus seemed that participants in our study distributed their attention across the ranks to a satisfactory degree (i.e., as opposed to just focusing on the first few results). This meant that if order effects were strong, we should have found an effect of search result rankings on attitude change.

Two other interesting metrics to look at were (1) the time participants spent exploring the SERP and (2) the number of URLs they clicked. These two metrics – that both contributed towards our *user engagement* measure (see Section 5.2) – could tell us more about participant's sincerity in doing the task as well as their motivation and behavior related to informing themselves on the debated topic. First, participants spent an average of 3.32 minutes on the SERP (sd = 1.89). Participants thus spent considerably more time here than the required minimum (i.e., two minutes). This indicated that participants took the task seriously and were motivated to inform themselves on the debated topic. Second, participants clicked a mean of 2.34 URLs (sd = 1.49) and 36% of them clicked the "more" button at the bottom of the SERP. The participants who clicked the "more" button did not, however, engage more in terms of their click behavior (mean = 2.45). These findings suggest that participants used the search result titles and snippets to obtain an overview of the topic and then clicked on particular URLs that interested them.

Overall, 70% of participants expressed an attitude change after viewing the SERP. That is, they moved at least one point on the Likert scale in their post-search attitude compared to their pre-existing

attitude towards their assigned debated topic. Mean absolute attitude change (over all conditions) was 1.06 (sd = 0.89), with 30% of participants experiencing an attitude change of two points or more on the Likert scale. This indicates that the search results we showed to participants had the potential to cause attitude change. In line with previous research [60], we find that most participants who reported attitude change (57%) became more *supportive*.

## 6.2 Hypothesis Tests

Table 3 shows the ANCOVA results. There was no significant difference between conditions (i.e., levels of ranking bias) in terms of attitude change ($F = 1.67$, $p = 0.19$, $\eta^2 = 0.01$; H1a; see Figure 2). We thus found no evidence in favor of **H1a**. In contrast, as a result of conducting a Bayesian ANOVA, we found moderate evidence in favor of the null hypothesis *opposing* **H1a**, namely that condition had *no influence* on attitude change ($BF_{01} = 8.56$).

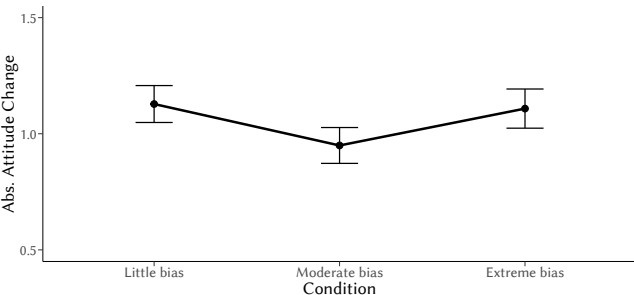

**Figure 2: Mean absolute attitude change over the three conditions. The error bars represent the standard error.**

**Table 3: ANCOVA (absolute attitude change as dependent variable). Colons represent interaction effects.**

| Hyp. | Variables | $F$ | $p$ | $\eta^2$ |
|------|-----------|-----|-----|----------|
| H1a | condition | 1.67 | 0.19 | 0.01 |
| H1b | condition:topic | 0.74 | 0.66 | 0.01 |
| H2a | AOT | 0.90 | 0.34 | 0.00 |
| H2b | user engagement | 0.01 | 0.94 | 0.00 |
| H3a | condition:AOT | 0.23 | 0.80 | 0.00 |
| H3b | condition:user eng. | 3.81 | 0.02 | 0.02 |
| H3c | condition:perc. div. | 2.93 | 0.06 | 0.01 |

The ANCOVA also revealed no direct effects of AOT ($F = 0.90$, $p = 0.34$, $\eta^2 = 0.00$; **H2a**) or user engagement ($F = 0.01$, $p = 0.94$, $\eta^2 = 0.00$; **H2b**). Similarly, there were no significant interaction effects between *condition* and *topic* ($F = 0.74$, $p = 0.66$, $\eta^2 = 0.01$; **H1b**), *AOT* ($F = 0.23$, $p = 0.80$, $\eta^2 = 0.00$; **H3a**), *user engagement* ($F = 3.81$, $p = 0.02$, $\eta^2 = 0.02$; **H3b**), or *perceived diversity* ($F = 2.93$, $p = 0.06$, $\eta^2 = 0.01$; **H3c**). We did not perform a mediation analysis to test **H2c** because we did not find *AOT* to be significantly related to *attitude change* (i.e., there was no effect to be mediated.)

The One-Way ANOVA showed no significant effect of *condition* on *perceived diversity* ($F = 0.07$, $p = 0.94$, $\eta^2 = 0.00$; **H4a**). Conversely, a Bayesian ANOVA revealed strong evidence in favor of the

opposing null hypothesis ($BF_{01}$ = 31.23), indicating that participants did *not* perceive different levels of diversity.

In sum, we cannot reject any of the null hypotheses opposing the hypotheses we specified in Section 2. Bayesian analyses reveal moderate to strong evidence that there are no effects of search result rankings on user attitudes and perceived diversity here.

## 6.3 Exploratory Findings

Our results suggest that although most users experienced attitude change due to viewing SERPs on debated topics, attitudes were not affected by top 10 search result rankings or individual differences that we measured. We aim to further understand these findings in this subsection and first probe the data for order effects that may not have been picked up by our previous analyses. However, if there were no order effects, what else drove participants' attitude change? We discuss the roles of exposure effects, confirmation bias, and position bias. Note that the statistical analyses presented here were *not* pre-registered: they are of exploratory nature.

*6.3.1 A Closer Look at Order Effects.* Before turning to alternative explanations for attitude change in our study, we examined the order effects-hypothesis more closely. A total of 77 users in our study consumed exactly as many opposing as supporting documents. At least these users should have changed their attitude in accordance with higher-ranked results if order effects occurred. However, we found no difference in attitude change between users in this group who saw opposing-biased SERPs and those who saw supporting-biased SERPs ($t = -0.20$, $df = 75$, $p = 0.85$).

*6.3.2 Exposure Effects.* If order effects are not responsible for SEME, exposure effects may play a bigger role in search results-driven attitude change than previously expected. Exposure effects suggest that the higher the consumed proportion of documents pertaining to a particular viewpoint, the stronger the tendency to adopt that viewpoint. Our study aimed to mitigate exposure effects by letting users explore viewpoint-balanced SERPs for a minimum of two minutes. Nevertheless, some users consumed much higher proportions of supporting or opposing documents than others. We indeed found a relationship between the proportion of supporting documents among the results a user clicked on and attitude change ($r = 0.34$, $df = 342$, $p < 0.001$; result of a Pearson correlation analysis).[15] As exposure effects would predict, participants thus changed their attitude in accordance with the documents they consumed.

*6.3.3 No Evidence for Confirmation Bias.* Previous research has demonstrated that *confirmation bias* (i.e., a tendency to engage with proattitudinal information) can occur in web search [31, 65]. An explanation for attitude change in our study could thus be that users engaged with mainly proattitudinal search results, which subsequently caused the exposure effects. However, we found no evidence for a difference between pre-existing attitudes (i.e., somewhat opposing, neutral, somewhat supporting) concerning the proportion of supporting documents that participants clicked on ($F = 0.01$, $p = 0.93$, $\eta^2 = 0.00$, result of a one-way ANOVA).[16]

---

[15]We did not conduct the same analysis for the proportion of opposing results due to symmetry. Twenty participants were excluded from this analysis because they did not click on any URLs (i.e., no proportion of supporting documents could be calculated).
[16]See Footnote 15.

*6.3.4 When Position Bias Meets Ranking Bias.* We show in Section 6.1 that, despite our efforts to make users engage with the full SERP, there was some position bias: click proportions decreased from the first to the sixth result (i.e., from 0.55 to 0.15). This means that in the *extreme bias* condition, the average user consumed documents that largely promoted one particular viewpoint because the first five results all pertained to the same viewpoint in this condition (see Table 4). If exposure effects took place, there should thus be a difference in attitude change between users that saw the supporting-biased SERPs and those who saw the opposing-biased SERPs. We looked at attitude change per condition, split by the viewpoint bias on the SERPs (see Table 5). We indeed observed a tendency for values of attitude change to drift apart as conditions became more extreme. Here, the *directions* that these values drifted towards corresponded to the direction of the viewpoint bias on the SERP (e.g., attitude change was more positive in more extreme conditions when the SERP was supporting-biased). A t-test comparing attitude change between the two bias directions in the *extreme bias* condition revealed a potential difference ($t = 2.61$, $df = 113$, $p = 0.01$).

**Table 4: Proportions of supporting documents among the search results that users clicked on (± std. dev.) in each condition, split by advantaged viewpoint on the SERP.**

| Condition | Prop. of supporting documents among clicked results | |
| | Supporting-biased SERP | Opposing-biased SERP |
| --- | --- | --- |
| Little bias | 0.53 (±0.30) | 0.49 (±0.34) |
| Moderate bias | 0.66 (±0.33) | 0.44 (±0.33) |
| Extreme bias | 0.74 (±0.30) | 0.32 (±0.37) |

**Table 5: Mean attitude change (± std. dev.) in each condition, split by advantaged viewpoint on the SERP.**

| Condition | Mean attitude change | |
| | Supporting-biased SERP | Opposing-biased SERP |
| --- | --- | --- |
| Little bias | 0.13 (±1.37) | 0.11 (±1.5) |
| Moderate bias | 0.21 (±1.34) | 0.04 (±1.18) |
| Extreme bias | 0.50 (±1.30) | -0.17 (±1.50) |

## 7 DISCUSSION

We expected to find that user behavior is guided by order effects, and predicted that *changing the order* of items on an overall viewpoint-balanced SERP would lead to varying degrees of attitude change. However, we found *no evidence for order effects*; conversely, we found moderate evidence that there is no effect of search result order on attitude change in this context (i.e., a conclusion drawn from a Bayesian analysis that quantified evidence in favor of the null hypothesis; **RQ1**). Our results further do not contain evidence for an interaction effect of topic and the order of search results. We similarly found no evidence for direct effects (**RQ2**) or interaction effects (**RQ3**) concerning other factors we measured (i.e., AOT, user engagement, and perceived diversity). Moreover, our results suggest that participants did not perceive the varying degrees of viewpoint diversity (i.e., different levels of ranking bias) in the SERP we presented to them (**RQ4**). Our findings therefore imply that order effects – if they exist in this context – contribute less strongly to SEME than one might anticipate.

## 7.1 Explaining SEME?

Exploratory analyses that we conducted indicate that exposure effects as a result of viewing search results may cause attitude change. As exposure effects predict, we found that the more search results pertaining to a particular viewpoint users consumed, the more they tended to adopt that viewpoint. Our results suggest that users did not have confirmation bias when engaging with search results but instead selected documents with a position bias (i.e., they were likely to consume higher-ranked results). This selection then led some users to engage with more documents pertaining to a particular viewpoint, which in turn guided their attitude change.

How do all these results fit together? If participants were affected by position bias in selecting documents and exposure effects regarding their attitudes, why did this not result in different levels of attitude change across conditions in our study? A potential explanation is that our manipulation (i.e., presenting overall viewpoint-balanced but ranking-biased SERPs) was too weak for SEME to occur. Previous studies that investigated SEME exposed users to SERPs where one viewpoint was in the majority [3, 15, 47, 60]. This allowed for much more reliable exposure effects as most users would have consumed a great proportion of one particular viewpoint.

It should be pointed out that many different cognitive biases and other external factors play a role in web search [6]. Our study highlights that explaining SEME is a complex problem that requires at least a thorough understanding of (1) how users select documents from SERPs when searching for debated topics and (2) how the selected results affect them. After several studies have shown contexts in which SEME can occur [3, 15, 47, 60], we show that it does not occur in all cases of viewpoint-related ranking bias. Our results suggest that users may not exhibit strong order effects when consuming search results but that exposure effects can contribute to attitude change as a result of viewing search results.

## 7.2 Implications

Our findings have implications for the measurement and mitigation of ranking bias and SEME. First, if order effects do not contribute to SEME, the top-$k$ portion of the ranking does not need to reflect optimal viewpoint diversity at every rank $k$. This means that the discount function in ranking bias metrics should be chosen according to a good estimate of at which ranks a lack of diversity could cause SEME. For example, it might be suitable to apply the log-discount in steps of ten [63] or to apply an alternative discount function [53].

Second, if exposure effects are the main contributor to SEME, it seems plausible that it can (in part) be mitigated by addressing the ranking bias so that there is a viewpoint balance on the first SERP. Several re-ranking algorithms have already been proposed for similar purposes [5, 8, 10, 39, 67].

Third, applying an (interface) intervention that makes users consider a more diverse selection of documents could also mitigate SEME. Previous research has already investigated this option and found that SEME could be mitigated by alerting users to an existing ranking bias [16]. This alert led users to examine more (and thereby a more viewpoint-balanced set of) search results. Similarly, interventions that nudge users to engage with more search results (e.g., by displaying search results in a different format than a list [28]), increase cognitive reasoning [45], provide additional information

about the search topic or the ranking [36, 62, 64], visualize bias among search results [11], or recommend counter-attitudinal substitutes for selected documents [12, 65] could prove fruitful here.

## 7.3 Caveats, Limitations, and Future Work

Our future work will investigate exposure effects in the context of web search on debated topics in more detail and further develop an understanding of their relationship with ranking bias metrics. Note that we only measured attitude change twice (before and after users interacted with SERPs) and did not collect data on the order in which users clicked on the different documents they engaged with. We thereby cannot deduce the *point at which* attitude change occurred. Furthermore, we cannot ascertain whether phenomena such as confirmation bias affect users on a more nuanced level; e.g., only early or late in the search. Examining the exact dynamics of attitude change in this context presents an exciting challenge for future research. We also only investigated user behavior in exploring single SERPs in single search sessions that lasted a minimum amount of time (i.e., two minutes). In our future work, we will explore how attitude change occurs in more realistic scenarios (i.e., providing users with deeper lists of search results whilst allowing for shorter exploration times and multiple search sessions). Participants' distribution over topics in our study was not balanced, which might have affected the results. Moreover, asking users to self-report their attitudes towards debated topics could have prompted them to evaluate their attitude; a process that otherwise might not have taken place. Future work could look into measuring attitude (change) in a more subtle and implicit fashion to rule out such effects.

## 8 CONCLUSIONS

We presented a user study investigating the effect of search result rankings on user attitudes. We found that viewing a viewpoint-balanced SERP containing ten search results related to a debated topic led to attitude change in a majority of users. However, neither the *order* in which these search results were ranked, nor the individual differences we measured affected attitude change. These findings imply that order effects are not a likely explanation for SEME in users with mild pre-existing attitudes. Instead, our exploratory analyses suggest that exposure effects could be responsible in this context (i.e., users adopting the majority viewpoint among the results they examine). We propose that simple interventions merit further study as user bias mitigation strategies.

## ACKNOWLEDGMENTS

This activity is financed by IBM and the Allowance for Top Consortia for Knowledge and Innovation (TKI's) of the Dutch ministry of economic affairs.

We also thank Alexandra Sarafoglou, Alisa Rieger, Shabnam Najafian, Oana Inel, Francesco Barile, and Rishav Hada for their comments on an earlier draft of this paper.

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
