# OpenReview forum: "This Is Not What We Ordered: Exploring Why Biased Search Result Rankings Affect User Attitudes on Debated Topics"
_ACM.org/SIGIR/Badging_

### Official Review · ~Johannes_Kiesel1 · 2021-07-02
**Approved**

**Comment:**

As discussed in the comments, this resource now fulfills the requirements for the requested badges.

I won't repeat the things I checked here.

**Awarded Badges:**

["Artifacts Evaluated – Functional", "Artifacts Evaluated – Reusable and Available"]

---

### Official Review · ~Jiqun_Liu2 · 2021-08-19
**Approved**

**Comment:**

After double-checking the resource items, I am happy to approve the requested badges.


**Awarded Badges:**

["Artifacts Evaluated – Functional", "Artifacts Evaluated – Reusable and Available"]

---

### Official Review · Program_Chairs · 2021-08-23
**Functional, Reusable, and Available Badges**

**Comment:**

According to the discussion you had with the reviewers and to the final reviews, your artifact is ready for badging.

We are happy to award you the following badges:
* Artifacts Evaluated – Functional
* Artifacts Evaluated – Reusable and Available

To have your artifact included in the ACM DL we need to received a single zip file containing it.

Could you, please, send it to us at the following email address:

aec_sigir@acm.org

at your earliest convenience?

**Awarded Badges:**

["Artifacts Evaluated – Functional", "Artifacts Evaluated – Reusable and Available"]

---

### Public Comment · ~Johannes_Kiesel1 · 2021-06-15
**Current status**

Dear Tim, Nava, Ujwal, Alessandro, and Benjamin,
Thank you for your submission!

Please find the current state of my review below (issues marked with "(!)"). As of now, I would award you "Artifacts Evaluated – Functional", but not yet "Artifacts Evaluated – Reusable and Available".

Feel free to discuss here or tell me if you think you addressed the issues.


Awarding Artifacts Evaluated – Functional
-----------------------------------------
The artifacts associated with the research are found to be:
Documented: YES
  + Some README files exist
  + Comments in code
Consistent: YES
  + 280 search result items: Data Set/Search results/search_results_annotated.csv
  + 5 topics: Data Set/Search results/queries.csv
  + 364 participants: Results/Data/studyResults.csv
Complete: YES
  + Pre-registration plan
  + Search results (URL, title, snippet, ...)
  + Screenshots to document the study interface
  + Documentation on metric computation
  + Collected data
  + Code to produce results
Exercisable (runs without errors): YES
  + Data Set/Preliminary study/analyse_prelStudy.R
  + Results/Analyses/analyzeResults.R
Include appropriate evidence of verification and validation: YES
  + No obvious problems


Awarding Artifacts Evaluated – Reusable and Available
-----------------------------------------------------
Very carefully documented: PARTIALLY
  - (!) No main README and no README for most directories: several data files without documentation
  - (!) No code setup guide: in this case not a problem for me (knowing a bit of R), but maybe for others
  - (!) Partially unclear purpose of code (probably alleviated through central README): I needed a bit to figure out the purpose of "Data Set/Preliminary study/analyse_prelStudy.R"
  - (!) Linking output of Results/Analyses/analyzeResults.R to the paper tables is not trivial
  + Main CSV file has a README: Results/Data/README.md
  - (!) No documentation (at least I found none) how to use Results/Analyses/bayesianAnalyses.jasp
Well-structured to the extent that reuse and repurposing are facilitated: PARTIALLY
  + Well-structured CSV files
  - (!) One jasp file: some archive that requires specific program. Can it (also) be provided in a text-based format?
Associated artifacts have been made permanently available for retrieval: YES
  + Hosted on osf.io

---

> ### Public Comment · ~Tim_Draws1 · 2021-06-21
> **Updates**
>
> Dear Johannes,
>
> Thank you for the careful evaluation of our repository. I believe that we have now fixed all the issues you mention regarding "Artifacts Evaluated – Reusable and Available". Here is an overview:
>
> (!) No main README and no README for most directories: several data files without documentation
> ---> Added README's everywhere (aside from the "Plots" subdirectory, which I thought was self-explanatory given the readme for the directory above)
>
> (!) No code setup guide: in this case not a problem for me (knowing a bit of R), but maybe for others
> ---> included instructions on how to run R codes and JASP files in the respective readme files
>
> (!) Partially unclear purpose of code (probably alleviated through central README): I needed a bit to figure out the purpose of "Data Set/Preliminary study/analyse_prelStudy.R"
> ---> indeed included this in the relevant readme file, I hope it is clear now
>
> (!) Linking output of Results/Analyses/analyzeResults.R to the paper tables is not trivial
> ---> added references to the tables in the paper to the R code
>
> (!) No documentation (at least I found none) how to use Results/Analyses/bayesianAnalyses.jasp Well-structured to the extent that reuse and repurposing are facilitated: PARTIALLY
> ---> added this to readme, see comment above
>
> (!) One jasp file: some archive that requires specific program. Can it (also) be provided in a text-based format? Associated artifacts have been made permanently available for retrieval: YES
> ---> unfortunately the JASP file does need the JASP software to be run on one's own computer; however, it can be fully viewed in the browser on the OSF by just clicking on it (added this hint to readme as well)
>
> I hope that we have addressed the above issues sufficiently to earn the second badge, too! If not, please let us know and we will fix remaining issues asap.
>
> Kind Regards,
> Tim

---

> > ### Public Comment · ~Johannes_Kiesel1 · 2021-07-02
> > **Issues Resolved**
> >
> > Dear Tim,
> >
> > I'm sorry that it took me so long. I now checked on your updates and I think they are totally sufficient.
> >
> > And congratulations on the paper!